# An alternative pattern of head expansion during feeding in cichlids
Jana De Ridder [1,2], Vincent Dujardin[1], Julia Camacho Garcia [3], Wilson Sawasawa [3], Peter Aerts [1], Hannes Svardal [3,4] & Sam Van Wassenbergh [1] ✉

Specialized feeding methods evolved repeatedly from a suction-feeding strategy in cichlids. How algae-eaters altered their suction mechanics to transport detached algae efficiently, and how this may hinder capturing larger prey, remains unclear. Here, we study the kinematics and time-resolved volumetrics of a piscivore, an algae picker/nibbler, and an algae scraper from Lake Malawi feeding on attached algae tablets and free pieces of shrimp. Algae specialists lack the common anterior-to-posterior expansion and compression waves of the head, instead exhibiting a synchronous expansion pattern. This alternative pattern may allow algae retention due to low-amplitude dilation of the gill rakers, maximized suction flow speeds for a given local expansion amplitude, and rapid sequences of suction cycles. The trade-off between powerful suction and efficient feeding on algae may explain why algae scrapers' opportunistic switching to suction feeding can only be successful on easy prey, and may have impacted cichlid diversification and trophic niche constraining.

Understanding functional trade-offs plays a central role in organismal and evolutionary biology[1]. These trade-offs derive from conflicting functional demands, implying that adaptations for one specific task can lead to reduced performance in another. They can drive the evolution of specialization in which species occupy narrow ecological niches where they excel[2,3]. Specialization is common in adaptive radiations, where ancestral species diversify into various forms, each adapted to a different ecological role[4,5]. However, the trade-offs accompanying such specialization can also limit a species' ability to adapt to new environments or shifts in ecological conditions[6]. Therefore, identifying conflicting demands on organisms' morphology, physiology, mechanics, or behaviour is crucial for understanding ecological niches and evolutionary processes.

An important model system in the study of the role of functional trade-offs in the evolution of an adaptive radiation are cichlids[7–10]. Cichlids are widely distributed, but the most species-rich assemblages are found in the African Great Lakes[4,11]. In these lakes, a high diversity in food preferences and trophic morphologies are found in co-existing cichlid species. How cichlids came to be so diverse, for example in Lake Malawi, is a long active area of research, and multiple contributing factors have been identified[12,13]. The decoupling of the oral and pharyngeal jaws, for example, is thought to have facilitated feeding mode diversification[9,14], but see Conith et al.[15].

Competition for food resources is generally considered an important factor in the emergence of specialized feeding modes and morphologies in African Great Lake cichlids[4,12,16–19]. However, several studies have questioned this hypothesis. The lab experiments of Liem[20] were an important start of the discussion. Species presumed to be specialized in scraping algae were readily consuming other kinds of food[20]. Also in Lake Tanganyika, presumed trophic specialists were observed to leave their foraging grounds to temporarily feed on a school of clupeids passing by[21,22]. The apparent contradiction of species with extreme phenotypic specialization displaying even a wider feeding repertoire than species assumed to be generalist feeders based on their morphology became later known as 'Liem's paradox'. Robinson & Wilson[23] explained this phenomenon by dividing food into two groups: (1) preferred and intrinsically easy to use, and (2) nonpreferred and difficult to use. If the preferred food is abundant, fish specialized in other food types will also consume it, while specialization is relevant in periods of scarcity of the preferred food when fish need to fall back to more difficult to process food[23].

Success in switching between food types will depend on the functional trade-offs to capture and process both the preferred and fallback foods efficiently[23]. To capture food underwater, two strategies can be generally be discerned, namely suction and biting[24,25]. The latter category includes the scraping or tearing of rock algae, a fallback food in many cichlid species. Suction feeding uses waterflow to engulf food, while biting relies on physical contact with jaws and teeth. These two mechanisms are assumed to have conflicting morphological demands[24,26–28], but the complex cranial system of fishes allows them to be combined[7,20,24,25,29,30]. Evolutionary shifts in predominant usage of the two mechanisms are also common in cichlids[31].

[1]Laboratory of Functional Morphology, Department of Biology, University of Antwerp, Antwerp, Belgium. [2]Evolutionary Morphology of Vertebrates, Department of Biology, Ghent University, Gent, Belgium. [3]Evolutionary Ecology Group, Department of Biology, University of Antwerp, Antwerp, Belgium. [4]Vertebrate Evolution, Development and Ecology, Naturalis Biodiversity Centre, Leiden, The Netherlands. ✉e-mail: sam.vanwassenbergh@uantwerpen.be

Compared to the many studies that focussed on trade-offs in oral jaw function[7,10,32,33], relatively little is known on how the suction-feeding apparatus is affected by specializations for algae feeding, but see refs. 25,34,35. Some suction capacity must inevitably be retained even in the most specialized algae scrapers because it is essential to transport the detached algae to the back of the buccopharyngeal cavity for swallowing[20], and because respiration also relies on suction and pumping of water through the buccopharyngeal or 'mouth' cavity[36]. A recent study compared the 2D mouth cavity expansion characteristics based on lateral-view high-speed videos in 56 species of cichlids, and found clear differences in feeding motion in species with different diets[37]. The most conspicuous difference was the higher overall cranial kinesis in species that eat large, mobile prey. In addition to these differences in cranial expansion magnitude, it was also hypothesized that suction kinematics in these predatory species is more efficient, but this immediately led to the question why diet of slow or immobile prey would result in a less efficient way of suction generation[37].

Gaining a better insight into suction trade-offs and efficiency of suction on algae versus mobile prey calls for a detailed comparative analysis of the kinematics of suction feeding in cichlids with different diets. Previous comparative studies[34,37,38] investigated suction-feeding kinematics in two dimensions based on lateral-view videos. However, including the third dimension is crucial for analysing the volumetric expansion of the mouth cavity during suction. Here, we analysed the three-dimensional kinematics of the head of three species of Malawi cichlids with different morphologies and feeding behaviours (Fig. 1): (1) *Rhamphochromis* sp. 'Chilingali', a piscivore with elongated body and jaws[39] living in shallow waters to feed on small fish[31,40] (Figs. 1A), (2) *Chindongo saulosi* an algae picker/nibbler with a small mouth living close to the rocks to eat loose-hanging algae from rocks[40] (Fig. 1B), and (3) *Labeotropheus trewavasae*, an benthopelagic algae scraper with a broad, straight mouth and fleshy overhanging nose which eats algae from mostly vertical rock surfaces[40] (Fig. 1C). In this study, we evaluated

whether and how mouth cavity expansion in 3D is different in predatory versus algae-eating cichlids. From this we will argue how the cranial expansion system may be functionally tuned to this type of food, and how modifications may adversely affect the fish's capacity for powerful suction generation as observed in piscivores.

## Results
### Morphology
The three species studied, for which the results will be reported in order of increasing specialization in algae feeding, differed in head shape and skeletal morphology (Fig. 1). The most notable morphological differences were evident in the head's width-to-length aspect ratio (maximum head width divided by jaw-tip-to-opercular-tip length), which increased from 0.36 in *R.* sp. 'Chilingali' (Fig. 1A), to 0.61 in *C. saulosi* (Fig. 1B), and reached 0.67 in *L. trewavasae* (Fig. 1C). The orientation of the lower jaw rami (refer to dentary and articular labels in Fig. 1) from the symphysis to the quadrate joint, as viewed from a ventral perspective, increased from about 10° (Fig. 1A), to 30° (Fig. 1B), and culminated at 45 ° (Fig. 1C) among the species in our series. The angle between the ceratohyals and the midsagittal plane increased from approximately 15° (Fig. 1A), to 30° (Fig. 1B), and to 50° (Fig. 1C) across the three species. Oral tooth shape ranged from relatively long and sharp unicuspid teeth (Fig. 1A), to smaller and more densely packed bicuspid and tricuspid teeth (Fig.1B), and to multiple layers of tricuspid teeth on a mediolaterally straight jaw (Fig. 1C).

### Feeding behaviour
Based on observations during the filming period and reviewing the videos in slow-motion, qualitative descriptions can be made on the feeding behaviour of the studied species on the two experimental food types, namely attached algae tablets and free pieces of shrimp (see also Supplementary Movie 1). The piscivore species *R.* sp. 'Chilingali' initiated suction with the mouth

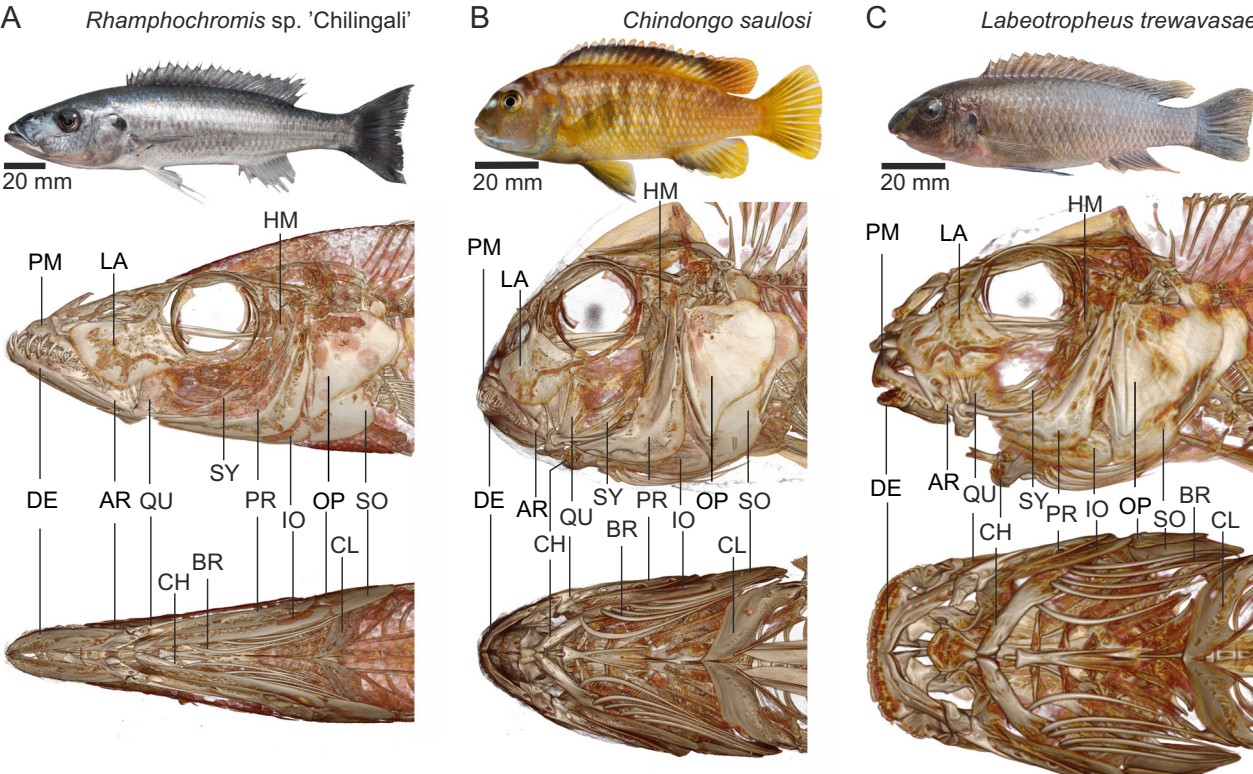

**Fig. 1 | Morphology of the three species studied.** Species are (**A**) the piscivore *Rhamphochromis* sp. 'Chilingali', (**B**) the algae picker/nibbler *Chindongo saulosi*, and (**C**) the algae scraper *Labeotropheus trewavasae*. Top panels show lateral-view photographs. Middle and bottom panels show CT-scan 3D-renderings of the bone skeleton from, respectively, lateral and ventral view. The indicated skeletal elements are: DE dentary, PM premaxilla, AR articular, CH ceratohyal, LA lacrymal, SY symplectic, PR preopercular, IO interopercular, SO subopercular, OP opercular, BR branchiostegal, HM hyomandibular, CL cleithrum.

close to the spirulina tablet. This sometimes sufficed to draw pieces of algae from of the tablet, but often the jaws were placed on the tablet or the tablet is scratched and some algae were detached when closing the mouth. This species usually performed one suction event, swam away, and repeated this later. In case the jaws grab onto the tablet, a second suction act follows immediately afterwards. The algae nibbler species *C. saulosi* and algae scraper *L. trewavasae* generally performed several successive bite-suction events before moving away from the spirulina tablet, and stayed close to the tablet after closing the mouth. In between events, the mouth closed completely. While *C. saulosi* generally feeds horizontally, *L. trewavasae* tends to feed on the vertically attached spirulina tablet in a nose-up body orientation. Videos with *L. trewavasae* feeding with an almost horizontal body position were selected to achieve sufficient image sharpness and aid the ventral-view image analysis. When feeding on the spherical piece of shrimp, *R.* sp. 'Chilingali' used a single suction act to engulf the food. In *C. saulosi* and *L. trewavasae*, however, the food item was sometimes grabbed with the jaws before opening the mouth again and sucking in the piece of shrimp.

## Anatomical landmark kinematics

The two food types (spirulina *vs* shrimp) evoked a number of significant differences in the kinematic variables (dashed *vs* full lines in Fig. 2) derived from the tracked head landmarks. The effect of food on kinematics generally did not depend on the species, as the interaction was not significant for any of the kinematic variables but one (maximal operculum-hind abduction; Supplementary Table 2). In general, expansion amplitudes were larger when feeding on the sinking piece of shrimp compared to feeding on the algae tablet. This was the case for mouth opening ($F_{1,\,112} = 72.221$, $p < 0.0001$), hyoid depression ($F_1 = 40.170$, $p < 0.0001$), premaxilla protrusion ($F_{1,\,111} = 16.468$, $p < 0.0001$), and suspensorium abduction ($F_{1,\,113} = 16.422$, $p < 0.0001$) (Supplementary Table 3). For operculum abduction a significant

increase was found for feeding on shrimp compared to spirulina for the front of the operculum ($F_{1,\,113} = 91.478$, $p < 0.0001$). This also applied to the hind of the operculum, but only for the piscivore *R.* sp. 'Chilingali' ($t_{108} = 6.341$, $p = 0.0001$; Table S4), whereas only a tendency ($0.1 < p < 0.05$) was found for algae picker *C. saulosi* ($t_{105} = 2.636$, $p = 0.0977$), and it was not significant for algae scraper *L. trewavasae* ($t_{111} = 0.453$, $p = 0.9975$). Food type also affected the timing of opercular abduction ($F_1 = 11.800$, $p = 0.0008$): maximal excursions were reached later when feeding on spirulina compared to feeding on shrimp. A complete overview of statistics results can be found in Supplementary Table 3. Individual variation within species is shown in Supplementary Fig. 2.

Feeding kinematics differed significantly between the three species (colours in Fig. 2; Supplementary Table 3). In general, the piscivore *R.* sp. 'Chilingali' had significantly larger normalized expansion amplitudes compared to the algae-eating species. Maximal mouth opening, for example, differed significantly between species ($F_{2,\,5.71} = 48.105$, $p = 0.0003$). The piscivore had a significantly larger mouth opening, the algae scraper *L. trewavasae* showed the smallest opening and algae picker *C. saulosi* lies in between these two species (Fig. 2A). However, the difference between the two algae feeders in maximal opening of the mouth was not significant ($t_{6.23} = -1.178$, $p = 0.5053$). Normalized hyoid depressions gave comparable results (Fig. 2C), with significant differences between species ($F_2 = 54.423$, $p < 0.0001$) and significantly larger hyoid depressions in the piscivore *R.* sp. 'Chilingali' compared to both algae-feeding species (Supplementary Table 3). Both algae feeders showed significant smaller abductions than the piscivore *R.* sp. 'Chilingali' for the suspensorium (*C. saulosi*: $t_{3.45} = -11.798$, $p = 0.0007$; *L. trewavasae*: $t_{10.2} = -13.315$, $p < 0.0001$), and front of the operculum (*C. saulosi*: $t_{4.89} = -8.440$, $p = 0.0010$; *L. trewavasae*: $t_{8.17} = -9.610$, $p < 0.0001$). The hind of the operculum also showed this pattern for feeding on shrimp, but only partly for feeding on spirulina (only

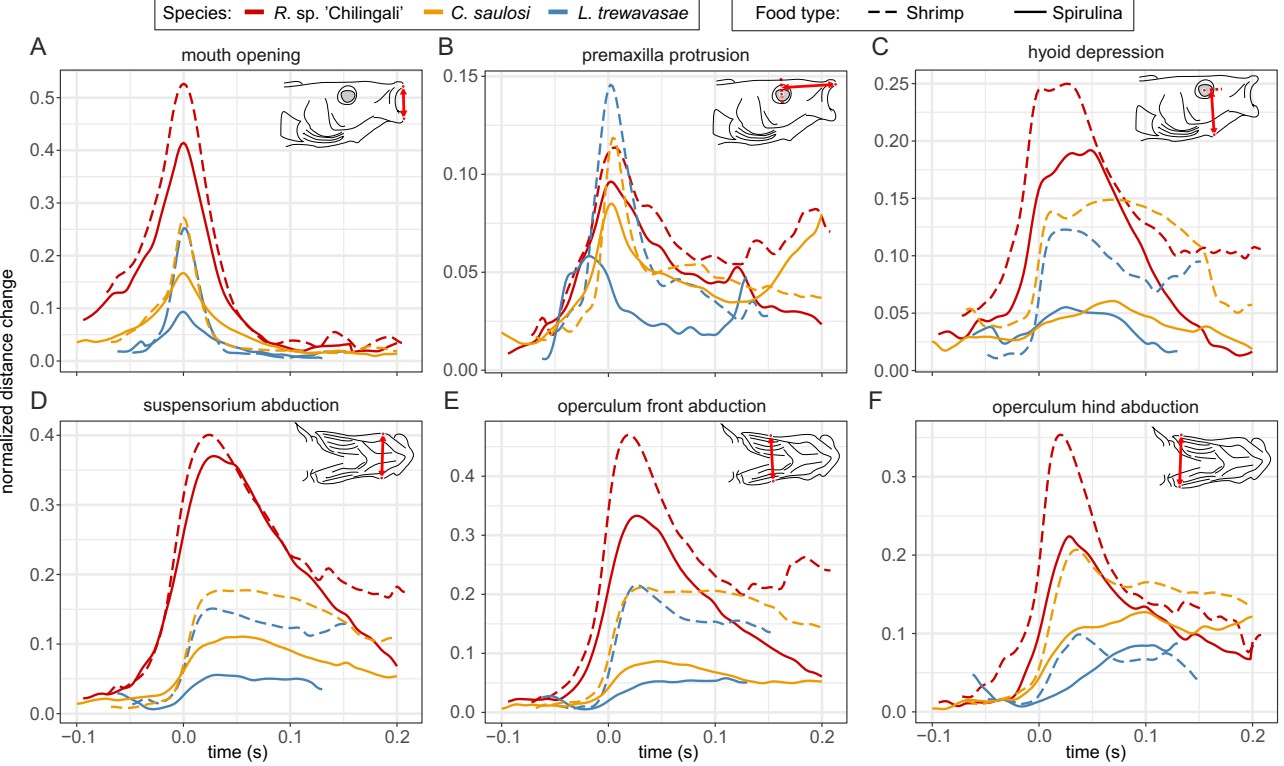

**Fig. 2 | Profiles of normalized distance changes of head landmarks.** Profiles are given for (**A**) mouth opening, (**B**) premaxilla protrusion, (**C**) hyoid depression, (**D**) suspensorium abduction, (**E**) operculum front abduction, and (**F**) operculum hind abduction. Distances are normalized using the cube root of the individual's head volume. Colours represent the different species, with piscivore *R.* sp. 'Chilingali' in red, algae picker *C. saulosi* in yellow, and algae scraper *L. trewavasae* in blue. Line

type represents the food type, with long dashes for shrimp and a solid line for spirulina. The averages are based on a varying number of trials (*R.* sp. 'Chilingali': $N_{Shrimp} = 25$ and $N_{Spirulina} = 26$; *C. saulosi*: $N_{Shrimp} = 21$ and $N_{Spirulina} = 23$; *L. trewavasae*: $N_{Shrimp} = 8$ and $N_{Spirulina} = 18$). See Supplementary Fig. 2 for inter-individual variability.

significant differences between *R.* sp. 'Chilingali' and *L. trewavasae*; Supplementary Table 3). A significant interaction between species and food type occurred for the abduction amplitude of this hind operculum landmark ($F_{2, 109} = 6.0931$, $p = 0.0031$). Premaxilla protrusion was not significantly different among species ($F_{2, 4.87} = 0.3285$, $p = 0.8267$).

A considerably larger plateau and slower post-peak decrease was observed in the buccal expansion profiles of the algae feeders (Fig. 2; Supplementary Fig. 2). This was the case for hyoid depression (Fig. 2C), suspensorium abduction (Fig. 2D), and operculum abduction (Fig. 2E-F). Both algae feeders differed from *R.* sp. 'Chilingali' when looking at the timing of reaching maximum suspensorium abduction. *C. saulosi* reached maximum abduction significantly later than the piscivore ($t_{117} = 4.108$, $p = 0.002$); *L. trewavasae* showed a similar tendency ($t_{117} = 2.220$, $p = 0.0721$). Also the timing of the maximal operculum abduction was significantly different in the piscivore compared to the algae feeders (compared to *L. trewavasae*: $t_{11.5} = 2.963$, $p = 0.0307$ for the front, and $t_{114} = 3.012$, $p = 0.0089$ for the back landmark; compared to *C. saulosi*: $t_{3.68} = 4.628$, $p = 0.0254$ for the front, and $t_{114} = 4.700$, $p < 0.0001$ for the back landmark).

### Head volumetrics

Expansion and subsequent compression of the head volume was observed with each 'bite' (Fig. 3; Supplementary Movie 1), but the magnitude differed between species and food type. The piscivore *R.* sp. 'Chilingali' had the largest magnitude in both feeding experiments. For all species, the volume increase was larger when they feed on shrimp. The largest relative increase in expansion magnitude from feeding on the algae tablet to feeding on the shrimp ball was observed in the algae scraper *L. trewavasae* (three times larger volume increase during shrimp vs algae feeding; Fig. 3C). For the other species the increase was almost two times larger when feeding on shrimp compared to algae (Fig. 3A,B). When comparing the spatio-temporal patterns of head expansion between feeding on the two food types for the piscivore *R.* sp. 'Chilingali', the pattern was comparable in shape, but the magnitude difference was particularly high at the back of the operculum (heatmaps in Fig. 3A).

Apart from the lower expansion amplitude, four other differences were observed when comparing the spatio-temporal patterns of head volume change between the two algae feeders *C. saulosi* and *L. trewavasae* and the piscivore *R.* sp. 'Chilingali' (heatmaps in Fig. 3): (1) the algae feeders showed a more gradual expansion phase with lower local expansion rates during algae feeding, whereby the maximum volume is reached later. (2) Their recovery phase of head compression was relatively slow (less dark blue colours on the heatmaps in Fig. 3B,C compared to Fig. 3A). (3) The spatio-temporal pattern differed: the piscivore showed an anterior-to-posterior progression of both expansion and compression (in other words, waves moving posteriorly along the head). In contrast, the main expansion and compression along the head was mostly synchronous in the algae feeders. This is notable as a vertical band of expansion and compression in the heatmaps of Fig. 3B-C, while this band was sloped in Fig. 3A. (4) An initial phase of compression was present before expansion starts in the algae feeders, which was not observed in the piscivore species.

### Discussion

A long-standing conundrum has been that morphological specialization can sometimes, unexpectedly, lead to increased dietary flexibility. This phenomenon, known as Liem's paradox, was demonstrated in cichlid fishes[20,21]. However, despite the impressive versatility of the feeding apparatus of many cichlids to handle a broad diet, biomechanical trade-offs will inevitably impact the performance and efficiency of feeding on different types of food. Gaining insight into the principles underlying eco-evolutionary trade-offs is essential to understand the selective pressures involved in niche specialization and thus the evolution and maintenance of biological diversity.

Here we focussed on potential trade-offs in the biomechanics of suction feeding. Our study was the first to analyse and compare the head's 3D expansion kinematics and volumetrics during suction generation across cichlid species with different dietary specializations. Three species were

compared with different levels of morphological specialization for feeding on algae (Fig. 1). All species showed the capacity to modulate their feeding kinematics in function of the food types offered in our experiment: faster and larger head expansions when feeding on the sinking piece of shrimp meat dropped from a tube compared to during bites at a disk containing compressed spirulina algae (Figs. 2,3). This result is in line with a large amount of literature showing fish's capacity to modulate feeding kinematics in response to food/prey characteristics like speed and size in virtually all species tested in this context[41–47]. In addition to this, and more importantly, several fundamental differences between the species were found in the biomechanics of suction generation. The potential adaptive value of these differences for feeding on algae, and implications for cichlid diversity and evolution are discussed below.

The magnitude of the expansion of the head was considerably lower in the algae feeders (Figs. 2,3). The pattern matched the degree of specialization for algae feeding with the 'algae picker' (*C. saulosi*) showing magnitudes that are intermediate between the piscivore (highest values in *R.* sp. 'Chilingali') and the algae scraper (lowest values in *L. trewavasae*). This means that the previously discovered link between jaw and hyoid depression magnitude, as viewed in 2D from lateral-view high-speed videos, and the level of evasiveness of the rift valley cichlid species' most common prey[37] can presumably be extrapolated to full 3D kinematics to also include lateral expansions of the head.

The less powerful and voluminous suction generated by fish that do not hunt evasive prey is generally described as suboptimal[26] or inefficient[37]. We suggest, however, that sequentially generating small buccopharyngeal expansions to suck in algae can be highly advantageous. To be able to ingest small food particles like algae into the fish's oesophagus, these particles must become concentrated near the oesophagus entrance. After being gulped into the mouth, these particles typically travel to the side of the buccopharynx where they are intercepted by the gill arches and rakers, which together form the branchial sieve[48–50]. Gathering of the accumulated food for transport and swallowing follows later. It is unlikely that the branchial sieve could maintain its filtration performance on small food particles if the branchial arches are being spread widely with each expansion of the head to suck in food. Hence, by limiting the expansion amplitudes, the sieving of algae may be optimized. Our observation that the buccal cavity keeps its expanded state for a longer time in algae feeders compared to piscivores (Fig. 2C-F; Fig. 3) fits this view: it may imply a reduced outflow speed and momentum of the water as it passes through the branchial slits and exits the opercular slits. Such reduced outflow speed and pressure are signs of flow across a filter with tight spaces. Additionally, the acceleration of large volumes of water for the uptake of small food particles would be energetically inefficient[51]. The observed, low-amplitude suction (or 'pump-filter feeding')[51], including prolonged expanded states in the algae-eating species, may thus be an adaptation to improve feeding efficiency on small food particles.

The implications of the observed variation in cranial expansion kinematics, along with the potential variations in the morphology of the buccal cavity, on the dynamics and patterns of water flow require further investigation. In the scope of the hypothesized trade-off between high-power suction and small-food suction, it would be informative to quantify the outflow properties as well, e.g. flow speed and momentum at the opercular exit. As mentioned above, the mechanical and morphological demands of these two tasks are vastly different. Quantifying the kinematics of the gill arches would also yield valuable insights. These arches are known to spread considerably after powerful suction[52] and may, therefore, compromise the functionality of small-food sieving. Several experimental approaches exist to gain such insights[49,53,54]. However, for Malawi cichlids, the generally small head size may be a limiting factor.

The algae-eating species showed a predominantly synchronous pattern of expansion of the head during feeding on both algae and shrimp (Fig. 3B,C). To our knowledge, this pattern has not been described before, and hence is deemed novel. It deviates from what is considered general and conservative among fish: the anterior-to-posterior (i.e., rostro-caudal) wave of head expansion[55–58]. The anterior-to-posterior wave of expansion implies

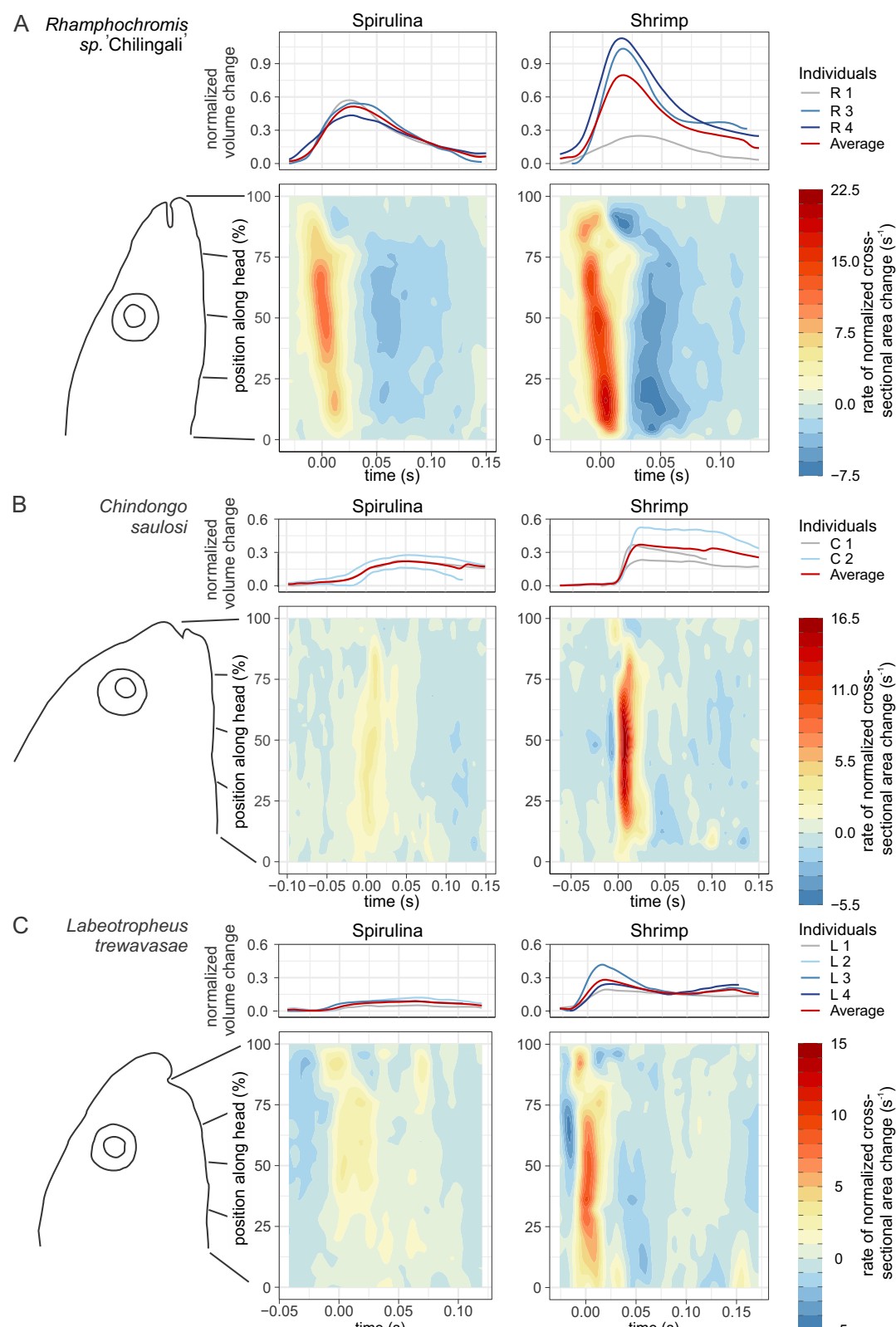

**Fig. 3 | Head volume change patterns.** Data are shown for (**A**) piscivore *R.* sp. 'Chilingali', (**B**) algae picker *C. saulosi*, and (**C**) algae scraper *L. trewavasae*, with feeding on the spirulina tablets on the left, and feeding on shrimp balls on the right. Each include, at the top, the total head volume changed normalized by the head's resting volume, and below, heat-map-type contour plots of the corresponding mean pattern of rate of normalized cross-section area change in function of time and position along the head (0% represents the back of the head at the height of the pectoral fin to 100% the front of the jaws; normalization by dividing by resting volume of the head to the power 2/3). Note that while the pattern of expansion (hot colours) in (**A**) is clearly sloped, corresponding to an anterior-to-posterior travelling wave of expansion, this pattern is largely vertical and hence a nearly synchronous expansion occurs in the algae eaters *C. saulosi* (**B**) and *L. trewavasae* (**C**).

that any given landmark on the head reaches its peak expansion speed, and usually also the peak excursion, slightly before the landmark immediately posterior to it[59,60]. It should be noted that also for the algae-eaters the mouth still opens earlier than the onset of the main expansion of the rest of the head, and a limited amount of 'flapping' of the opercular edge is observed after the peak expansion of the head, which corresponds to an anterior-to-posterior pattern (Supplementary Movie 1). However, as both the mouth opening and opercular edge flapping only negligibly contribute to head expansion, the overall expansion pattern can still be regarded as synchronous (Fig. 3).

Why would the algae feeders deviate from executing the typical and considered optimal anterior-to-posterior wave expansion? If local expansion amplitudes are constrained (as suggested above), the highest suction flow velocities at the mouth will be reached by expanding the entire head synchronously. It may therefore be the best compromise to exert sufficiently high suction forces on the algae without interfering with the branchial filtration process. An anterior-to-posterior wave of expansion, on the other hand, allows fish to better accelerate and sustain the movement of a parcel of food-containing water as it travels towards and along the buccopharyngeal cavity. Suction power, which may be a limiting factor for explosive suction feeding on evasive prey, can be used efficiently for prey capture by focussing the power output on the region around the prey (i.e. the wave region of high expansion speed and acceleration). For algae eaters, however, a high local water (and prey) momentum resulting from a well-coordinated expansion wave may even be detrimental for the retention of particles on the branchial sieve as such a high-momentum jet impacts the back of the buccopharyngeal cavity. Consequently, we hypothesize that the low-amplitude, synchronous expansion is an adaptive characteristic of pump-filter feeding fish like the algae-eating cichlids.

Synchronous expansions and compression cycles can also be completed quicker than wave-like expansions. The algae-eating cichlid species make use of repeated suction sequences when they are biting at the algae. Hence, the synchronicity in head expansion may be beneficial for the turnover time of the consecutive bite-suction cycles.

The perception of generality of the anterior-to-posterior expansion wave among fish may have been biased by the historical choices of model species. Most of the research thus far has been conducted on predatory species like trout[52], bluegill sunfish[61], largemouth bass[54], air-breathing catfish[30], or seahorses[62]. Fish that feed on plants or other particulate matter have been studied previously, like algae-scraping catfish[63] and cichlids[25,34,64], or carps feeding from the substrate[48,49]. However, no volumetric analyses of spatio-temporal patterns of head expansion have been performed in these studies. Consequently, it is possible that the observed synchronicity in head expansion is more common in non-predatory fish.

The incompatibility to switch between the observed wave-like (Fig. 3A) and synchronous (Fig. 3B-C) expansions presumably has an anatomical or constructional basis, but this needs further investigation. The narrow initial shape of the head and long hyoid and oral jaws of the predatory cichlids (Fig. 1A) may result in wave-like expansion because of the intrinsic mechanics[65,66]. In turn, the broader heads with short jaws and short sagittal length of the hyoid bone in algae eaters may be bound to unfold into a synchronous expansion. Musculoskeletal modelling of the jaw-hyoid-suspensorium-operculum complex would be useful to test this.

In conclusion, insights into the degree of incompatibility of functions helps us to better understand the underlying causes of trophic diversification[24] and constraints on evolution[67]. Our study sheds new light on the diet-specific selective pressures on the feeding system in fish, and therefore has implication on the evolution of cichlids in relation to feeding. Suction kinematics cannot be simply deemed either efficient or inefficient as it depends on the type of food that is considered. As discussed above, several aspects of the feeding kinematics of algae eaters are likely to facilitate an efficient uptake and retention of pieces of algae in suspension. It includes a newly discovered pattern of suction feeding in which head expansion is basically synchronous. The piscivorous fish in our sample seemed incapable of switching to this alternative pattern employed by the algae feeders (or vice versa) with the exception of reducing the expansion amplitude (Fig. 3).

Consequently, assuming that evolution towards high performance suction feeding is a one-way process leading to the powerful and voluminous suction in 'suction specialists' is therefore incomplete. Instead, we can conclude that the capacity for high-impulse suction trades off with the capacity for efficient suction of algae. This implies that adaptation to different diets in cichlids will necessitate morphological and biomechanical diversification of the cranial system beyond the jaws and teeth.

## Methods
### Specimens and experimental set-up
*R.* sp. 'Chilingali' individuals were offspring from a collection originating from Lake Chilingali, a satellite lake of Lake Malawi[68]. *L. trewavasae* and *C. saulosi* individuals were captive-bred individuals obtained from aquarium trade. Recordings were made from four *R.* sp. 'Chilingali', four *L. trewavasae*, and two *C. saulosi* individuals (individual head dimensions in Supplementary Table 1). For each species, a maximum of four individuals were kept together for four weeks in a 25 L aquarium enriched with an artificial plant and PVC pipes. Water temperature was kept constant at 25°C and the day/night cycle was maintained at 14 h/10 h with an LED light (Valoya LED Grow Lights, model L14: 14 Watt and NS12 spectrum). The aquarium had a relatively narrow extension (25 cm length x 8 cm width x 15 cm water height) where food was presented and high-speed videos captured (Fig. 4A).

Two food types were used. Firstly, a specialized herbivore feeding tablet containing 24% spirulina (Spirulina Tabs Nature, Sera, Heinsberg, Germany) attached on a vertical acrylic glass (polymethyl methacrylate) plate was used to study kinematics of feeding on algae (Fig. 4A). These tablets remain sufficiently firm to induce scraping for approximately ten minutes after being submerged, and were replaced if it took longer for the fish to start feeding. Secondly, a piece of meat of common shrimp (*Crangon crangon*) was rolled into 2-3 mm diameter balls and administered via a hollow plastic tube to study kinematics of suction feeding on a larger, moving food item. The sinking piece of shrimp was sucked up by the fish shortly after exiting the tube (Fig. 4A). The fish were filmed three times a week, always preceded by a fasting day to avoid satiation[69]. Food types were switched between video recording days and fish were fed spirulina on the non-recording days.

High-speed videos at 500 frames s$^{-1}$ and $1280 \times 1024$ pixels resolution were made from lateral and ventral views (Fig. 4A,B) on the fish using two X-PRI high-speed cameras (AOS Technologies AG, Baden Daettwil, Switzerland) with 55 mm f 3.5 Micro-Nikkor lenses (Nikon Corporation, Tokyo, Japan). A single trigger switch controlled both cameras, and hence ventral and lateral frames matched with a difference of less than 2 ms. Four infrared LED-arrays (18 W, 850 nm, 30° emission angle) provided illumination. For scaling, before every recording session or when the camera was repositioned, recordings were made with a transparent ruler held perpendicular to the lens axes and placed precisely at the position of the feeding fish.

### Morphology
To illustrate major skeletal differences in the feeding apparatus between the three species (Fig. 1), CT-scans were made (flexCT scanner, University Antwerp). Naturally deceased specimens from our collection were used for this purpose. Graphical 3D reconstructions were made using the 16-bit microCT presets in the volume rendering module of 3D Slicer (version 5.3.0)[70].

### Kinematic analysis
Landmarks on the head and pectoral region were tracked manually frame-by-frame using XMALab (Brown University, version 2.1.0). Those landmarks were, on the lateral views, the anterior tip of the lower (*Lj*) and upper (*Uj*) jaw, the middle of the eye (*Ey*), and the tip of the hyoid (*Hy*) (Fig. 2B). As a reference, a point on the pectoral fin (*Pf*), and a point on the neurocranium (*Nc*) were tracked as well. For both the pectoral fin and the neurocranium, the landmark was placed on a clearly visible spot such as a grey value shift or small bump on the fish, using approximately a spot at the same position for every individual. For the ventral views, the left and right margin of the fish were tracked for a landmark on the suspensorium (*Sl* and *Sr*), and

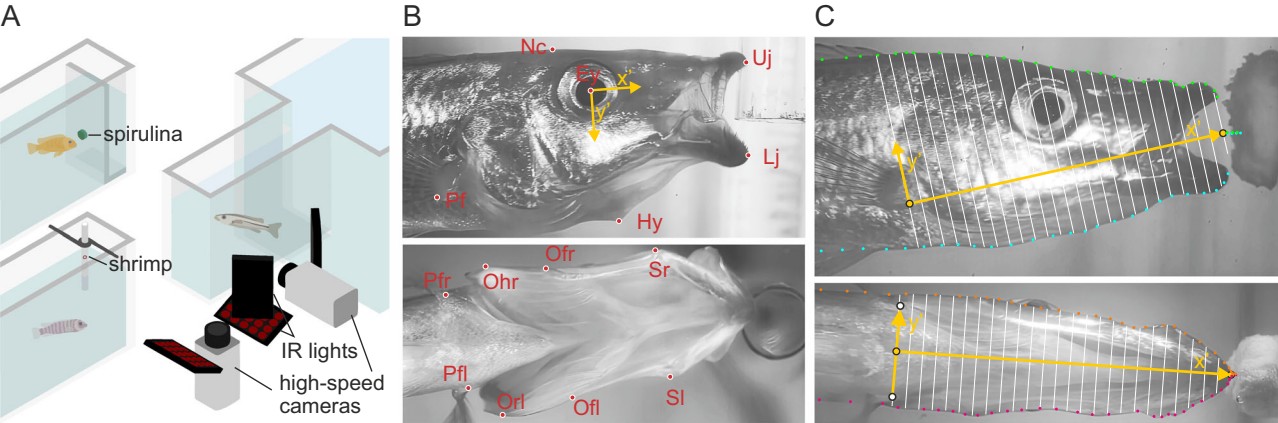

**Fig. 4 | Experimental set-up and extraction of kinematic data. A** Schematic illustration of the two foods offered (left), and camera and lighting set-up (right). **B** Landmarks tracked on lateral-view images (top): neurocranium (Nc), pectoral fin (Pf), eye (Ey), upper jaw (Uj), lower jaw (Lj), and hyoid (Hy). Landmarks tracked on ventral-view images (bottom): suspensorium left and right (Sl, Sr), operculum front (Ofl, Ofr), operculum hind (Ohl, Ohr), and pectoral fin at both sides (Pfl, Pfr). In yellow, the reference frame (x',y') moving with the fish is shown (eye = origin; x' direction = posterior-to-anterior). **C** Volumetric analysis using the ellipse method, showing the head contour landmark placement (green dots = dorsal; blue dots = ventral; orange dots = right; pink dots= left), the reference frame (x',y'; yellow) for section interval definitions and matching the lateral and ventral views, and the head heights and widths (white lines) that will define the cross-sectional ellipse axes.

a front (*Ofl* and *Ofr*) and a hind (*Ohl* and *Ohr*) point on the operculum (Fig. 4B). Those landmarks were again placed based on grey value or structure shifts. As a reference for the ventral views, the same landmark on the pectoral fin on both sides of the body and the front of the mouth was chosen.

Kinematic profiles of feeding were calculated from the coordinates of these landmarks. First, a moving axis was calculated for the lateral views using reference landmarks *Ey* and *Nc* (Fig. 4B). A constant angle was added to the *Ey-Nc* line orientation to obtain a posterior-to-anterior axis parallel to the line from the dorsal attachment of the pectoral fin to the mouth centre. The eye landmark *Ey* was used as the new origin, and *Uj* and *Hy* landmark coordinates were recalculated with respect to this moving axis. Profiles calculated were mouth opening (distance *Uj* to *Lj*), hyoid depression (perpendicular distance of *Hy* to the x' axis moving with the head reference markers; Fig. 2B), premaxilla protrusion (distance of *Uj* along the x' axis; Fig. 2B), suspensorium abduction (distance *Sl* to *Sr*), operculum front abduction (distance *Ofl* to *Ofr*) and operculum hind abduction (distance *Ohl* to *Ohr*). To compare kinematical profiles among individuals and species, a time reference point must be defined. The instant at which the gape is maximal was used as this 'zero-time'.

To take into account the size differences between species and individuals, a size weighing was done. To do so, calculated lengths were divided by the cube root of the volume of the head of each individual at rest (i.e. in unexpanded state), resulting in normalized distances and volume changes. To calculate the resting volume of each individual, the shape of the head was approximated as a half ellipsoid, since this shape generally closely resembles the head of a fish in an unexpanded state[71] (Supplementary Fig. 1). This volume was calculated for each individual as 2/3 π * head length * head width * head height (Supplementary Table 1). In comparison with a higher fidelity estimate of initial head volume based on the multi-frustum method (see further), we found a relatively small (4-14%) overestimation by the half-ellipsoid method (Supplementary Fig. 1). The trend was such that the differences between species would have marginally increased if we had used multi-frustum-based normalization.

Time-varying profiles for the six kinematics variables were filtered with a fourth-order zero phase Butterworth low-pass filter to reduce noise, and the average profiles of multiple feeding events per fish were determined. A cut-off frequency of 70 Hz was selected based on a visual inspection of the raw and filtered profiles. To prevent magnification of errors between calculations, intermediate variables, like the angles for the coordinate system transformations were also filtered in the same way.

The number of recordings differed between individuals and feeding experiment. For *R.* sp. 'Chilingali' a total of 25 trials for the shrimp and 26 for the spirulina experiment were made (number of recordings per individual with IDs in subscript): shrimp: $N_{R1} = 4$, $N_{R2} = 8$, $N_{R3} = 6$, $N_{R4} = 7$; spirulina: $N_{R1} = 7$, $N_{R2} = 6$, $N_{R3} = 7$, $N_{R4} = 6$), for *C. saulosi* 21 and 23 trials (shrimp: $N_{C1} = 11$, $N_{C2} = 10$; spirulina: $N_{C1} = 12$, $N_{C2} = 11$) were used, and for *L. trewavasae* 8 and 18 trials (shrimp: $N_{L1} = 2$, $N_{L2} = 2$, $N_{L3} = 3$, $N_{L4} = 1$; spirulina: $N_{L1} = 6$, $N_{L2} = 7$, $N_{L3} = 2$, $N_{L4} = 3$). Due to the longer head of *R.* sp. 'Chilingali', some of the most posterior landmarks were not in the camera's view during the whole feeding act. In those cases, these landmarks were tracked for a fraction of the time or not used for the corresponding parameters. This reduced the sample size for some parameters (to $N_{R2} = 5$ and $N_{R4} = 4$ for hyoid depression and premaxilla protrusion in the shrimp experiment and to $N_{R2} = 5$ and $N_{R4} = 4$ for operculum abduction hind in the spirulina experiment).

## Head volume quantification

By combining the ventral and lateral images, the local and instantaneous volume changes of the head of fish can be assessed by assuming that the cross-sections of the head can be closely approximated by ellipses[72]. For each species we selected the three best quality recordings from different individuals for which the complete head contour was visible. For every video frame, the dorsal, ventral, left and right contours of the head of the fish was digitized with XMALab software using 30 landmarks (Fig. 4C).

Reference landmarks were digitized and used to match the lateral and ventral views: the middle of the gape and the landmark at the pectoral fin for the lateral view, and the middle between the left and right landmarks at the pectoral fins and the front tip of the lower jaw (Fig. 4C). The contour coordinates were recalculated in a frame of reference moving with the head, based on the reference landmarks of the pectoral fin (or middle of both sides) and the mouth (yellow axes in Fig. 4C). The pectoral fin (center) landmark served as the origin, and the fin-to-mouth axis as the new x'axis. To correct for an underestimation of the total length of the head in 2D projection in ventral view due to the head being slightly pitched up or down, the ventral-view x'coordinates were scaled up so that the total length of the head matches that of the lateral view.

The length of the fish head at time t, $a_t$, was divided in 25 equally spaced cross-sectional slices with a thickness of $a_t/25$. The slices were modelled as a series of elliptical cylinders based on the local height and width of the head (Fig. 4C). These were measured as the distance (from dorsal to ventral, and from left to right) on fixed intervals along the length of the head using the

interpolation function FORECAST in Microsoft Excel. This procedure was repeated for each time step of the videos. After filtering height and width with the 70 Hz low-pass Butterworth filter, each slice with position index $i$ at time $t$ has an elliptical surface area $S_{i,t}$ of

$$S_{i,t} = \pi * \frac{h_{i,t}}{2} * \frac{w_{i,t}}{2},$$

with $h_{i,t}$ the local and instantaneous height, and $w_{i,t}$ the local and instantaneous width of the head. Local rates of cross-sectional area changes $\Delta S_{i,t}/\Delta T$ were then calculated by

$$\frac{\Delta Si, t}{\Delta T} = \frac{S_{i,t+1} - S_{i,t}}{(t+1) - t},$$

allowing to plot heatmaps of the rate of cross-sectional change over time in function of the position along the head of the fish as done by Michel et al.[73].

The total instantaneous head volume $V_t$ was calculated as

$$V_t = \sum_{i=1}^{25} \left( S_{i,t} * \frac{a_t}{25} \right).$$

The obtained temporal profiles of rate of cross-sectional area change and head volume were low-pass filtered with the abovementioned Butterworth filter.

## Statistics and reproducibility

Statistical analyses were done using R (version 4.0.3). To determine how suction-feeding differs between piscivorous and algae-eating cichlids, we tested the hypotheses that the three species differ in their cranial kinematics. Additionally, we also tested whether the two food types cause changes in kinematics, and whether the species respond differently to changes in food type (i.e., the interaction effect between 'food type' and 'species'). The analysis was performed on the anatomical landmark-based kinematic data (Fig. 4B). To allow such a statistical analysis, the continuous kinematic profiles were sampled at discrete time points. The variables extracted were, for each of the six kinematic profiles, the maximum value during the expansion phase, and the relative time point at which this maximum was reached. Linear mixed models with fixed effects of species and food type, and individual as random factor to account for the potential correlation between observations of the same individual were used (lmer function of lmerTest package, version 3.1.3). Interactions between food type and species were most of the times not significant, and therefore removed from the model. The effect of species and food type were tested using an $F$-test with Satterthwaite's approximation for the degrees of freedom (drop1 function using pbkrtest package, version 0.5.1). Next, pairwise comparisons between species and food type were carried out, adjusting the p-values for multiplicity using Tukey's method (emmeans function of emmeans package, version 1.7.4-1). In three models the variance of the random effect could not be estimated, due to a too low variability between individuals. In those cases, a linear model without a random effect was used (lm function), and the same pairwise comparisons were made from this simpler model. Assumptions of normality and homoscedasticity were evaluated visually, and no major deviations were observed. A double-sided critical significance level of $p = 0.05$ was used. The number of trials included in average kinematic profiles ($N$) is either mentioned in the figure legend (Fig. 2) or by a labelled indicating the number of included individual traces (Fig. 3). Statistical results report the degrees of freedom of the test. An overview of the sample size is given above at the end of the section 'kinematic analysis'.

## Ethics statement

The study received ethical approval from the University of Antwerp Ethical Committee for Animal Testing under file number ECD-2019-02. We have complied with all relevant ethical regulations for animal use. Required sample sizes were estimated beforehand based on previous studies on interspecific comparison of cichlid feeding kinematics[34,38,64].

## Reporting summary

Further information on research design is available in the Nature Portfolio Reporting Summary linked to this article.

## Data availability

All raw experimental data, namely the video materials used in this study, as well as the numerical data for the main figures are available from Figshare: https://doi.org/10.6084/m9.figshare.25612023[74].

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

## Acknowledgements
Thanks to Nemo Maes for helping with building the set-up. Thanks to Wendt Müller, Stefan Van Dongen, and Simon Baeckens for feedback on earlier versions. We thank University of Antwerp's DynXlab for help with CT-scanning. This research was funded by grant from the University of Antwerp's Special Research Fund to H.S. and S.V.W., and from the Research Foundation – Flanders to H.S. (grant number G047521N).

## Author contributions
Jana De Ridder: Investigation, Formal analysis, Visualization, Writing—Original Draft; Vincent Dujardin: Investigation, Formal analysis, Visualization; Julia Camacho Garcia: Supervision, Writing—Review & Editing; Wilson Sawasawa: Investigation; Peter Aerts: Conceptualization, Supervision, Writing—Review & Editing; Hannes Svardal: Conceptualization, Supervision, Resources, Writing—Review & Editing; Sam Van Wassenbergh: Conceptualization, Methodology, Resources, Software, Writing—Original Draft, Writing—Review & Editing, Funding acquisition.

## Competing interests
The authors declare no competing interests.
