## [Transparent Peer Review file · Communications Biology]

A novel pattern of head expansion during feeding in cichlids

Corresponding Author: Professor Sam Van Wassenbergh

Version 0:

Reviewer comments:

Reviewer #1

(Remarks to the Author)

This study investigates suction mechanics in three cichlid species from Lake Malawi differing in their head and mouth morphology, and foraging preferences. The authors measured mouth cavity expansion with volumetric analyses, suction capacity and three-dimensional kinematics, and provide qualitative descriptions of foraging behaviors.

The study finds interesting differences between piscivorous and alga-eaters in their feeding strategies, suggesting that the supposedly less efficient suction kinematics of algae feeders could have some benefits for catching smaller food particles, and the absence of waves of expansion and compression of the head while feeding in algae feeders, instead showing synchronous expansion patterns. AN interesting result is that all species show the capacity to modulate their feeding kinematics in function of food type.

Overall, I find the study interesting and well done.

Minor comments

-In the introduction and Figure 1 there are some morphological features of the three species analyzed, but this is not mentioned in the results and not really discussed. Either this is taken from a previous study, which I do not see clear, or should be more formally presented here.

-Related to the previous point, this study evaluates kinematic differences in feeding behavior, but it could be interesting to also see some discussion on the morphological differences in mouth and head morphology, other than volumes, that could relate to the preferred behaviors. One example could be mouth size and its relation to degree of mouth opening while suction. Another interesting aspect could be the elongation of the head and its relation to suction power.

Reviewer #2

(Remarks to the Author)

Summary

This study investigates the long-standing question of how the feeding apparatus of fish has contributed to (or restricted) their trophic and morphological diversity. Specifically, do specializations for biting or scraping create trade-offs with suction feeding performance? This is important for understanding how and why evolutionary shifts occur in adaptive radiations – of which African cichlids are a famous example.

The authors compare feeding in 3 African cichlid species (a piscivore, an algae picker, and an algae scraper) performing suction feeding (on shrimp) and biting/scraping (on algae). For each species and food type they measure the kinematics of dorsoventral and lateral mouth expansion and the change in cross-sectional area along the head and throughout the behaviour. The main conclusions are:

1. Algae eaters use a synchronous—rather than anterior-to-posterior—mouth expansion
2. Synchronous expansion is adaptive for algae eating
3. Synchronous expansion precludes powerful suction, forming a trade-off between powerful and efficient feeding for cichlid fishes and ultimately shaping trophic niches and diversification

Overall Impression

The first conclusion (synchronous expansion throughout the head in the two algae-eating species) is well-supported by the kinematic data, novel, and potentially transformative for studying the mechanics and ecology of fish feeding. Prior to this, an

anterior-to-posterior wave of expansion was considered a fundamental and universal element of suction feeding in fish. By demonstrating this alternative strategy for suction feeding, this study presents two key findings that have been largely overlooked in the field:

1. Slower, smaller-magnitude suction feeding is not necessarily “low performance”, but can be a successful strategy for consuming non-evasive food over longer durations. I agree wholeheartedly with the authors’ conclusion “Suction kinematics cannot be simply deemed either efficient or inefficient as it depends on the type of food that is considered” (p17, line 427). This raises interesting and important questions about when power—as opposed to work or efficiency—is important for suction feeding, and how muscles are employed for these different feeding strategies.

2. The kinematics of individual elements of the feeding apparatus need to be understood in the context of the volumetric expansion of the entire mouth cavity. I found it fascinating that an anterior-to-posterior wave of expansion could be interpreted from the linear kinematics of the two algae feeders, but not the measurements of mouth cross-sectional area. As the authors point out, this suggests synchronous expansion could be quite common but undetected in suction feeding fish (p17, lines 408-414).

I very much hope the field will embrace these findings and pursue the new avenues they open up.

The remaining two conclusions (on the adaptive advantages of algae-eating and the trade-off between powerful and efficient feeding) are plausible, but not strongly supported.

The authors made no measurements of feeding power, efficiency, or any type of performance; instead these conclusions are inferred from the kinematics (p7 line 83). But to my knowledge none of these can be accurately predicted solely from kinematics. For example, Van Wassenbergh et al., 2006 demonstrates the variable relationship between suction performance, volume, pressure, and morphology across different fishes.

The proposed explanations for how synchronous expansion is adaptive for algae eating are reasonable, but not tested or supported with any data. Can the authors refute the alternative hypothesis: synchronous expansion is sufficient for algae eating, but no more adaptive/advantageous than an anterior-to-posterior wave of expansion?

The piscivore’s capacity for powerful suction may not be the reason it is incapable of efficiently feeding on algae. First, we do not know how powerful or (in)efficient the piscivore actually is (see point above). This is a general problem in the field of how to compare feeding power across different species (e.g., Camp and Brainerd, 2022) and how to define a suction feeder as almost all fish use some combination of suction and ram (Longo et al., 2015). Second, even if we assume the piscivore is both powerful and inefficient at algae feeding, can the authors refute alternative hypotheses for its poor ability to feed on algae, such as its morphology or reliance on ram or body posture? For example, how might a powerful, piscivorous suction feeder with short-jaws and relatively little body ram—such as a bluegill sunfish—perform on algae?

The work is described in sufficient detail that it could be replicated, and I am pleased to see the authors have already prepared to make the video data openly available on FigShare. The statistical analysis seems appropriate to me, although this is not an area in which I am an expert.

Specific Comments

1. The impact of different head shapes, volumes, and volume calculation methods on the resulting expansion area and volume data needs to be discussed. As all the data are reported relative to resting head volume, this value is quite important. Please address the following questions in the methods or discussion, or provide references if this is already addressed in the literature.

A) Why was resting head volume calculated from only 3 measurements and modelling the head as a half ellipse (p9 lines 151-155)? It seems a more accurate measure could have been made from instantaneous head volume calculations at a frame before the onset of feeding (lines 174-206).

B) The two algae-eating species had broader and higher-volume heads; may this have impacted the normalised volume expansion? Could this method of measuring volume change from external head profiles introduce any bias when comparing species with different head shapes, e.g., the more slender piscivore and the more robust algae-eaters?

2. Personally, I feel a more detailed analysis of the interaction of kinematics and morphology, which is mentioned briefly (p17 lines 415-421), would add substantial value to this study. How much the different head shapes of these species could be impacting their kinematics and expansion? There are some really exciting questions here and the authors have already detailed 3D morphological data (Fig. 1). But I leave it to the authors to decide if such detail is within the scope of their study. It is a testament to the novelty and impact of their work that it already has me excited about new studies and questions!

3. Why wasn’t the maxilla tracked in ventral views—was it not clearly/consistently visible? Lateral abduction of the oral jaws seems like it is also contributing to mouth volume change, and getting this rostralmost region would help with tracking anterior-posterior timing. A measure of mouth width would have also allowed you to calculate the area of the mouth opening, which can be quite important for suction flows.

4. P6, line 50 – Do they really have conflicting morphological demands, given the content of the next two sentences? Consider re-wording this.

5. P6 lines 60-67; 82 – Please define what specific measurement is meant by ‘mouth expansion’ and is it the same this study (line 82) and Martinez et al 2018 (lines 60-67)?

6. P8 lines 139-147– Please explain how was the y-axis of the head reference frame defined? And how was the constant angle determined?

7. P11, line 237 – This is the first and only mention of pectoral girdle retraction: was it measured in any species? Did it occur in any species? Please provide more detail or remove this reference.

8. P12, lines 253-259 – I’m surprised that suspensorium abduction was significantly different between food types for the piscivore (as these lines look very similar in Fig. 3D)...but then hind operculum was NOT different between food types for the picker (as these lines look quite different in Fig. 3F). Have I misinterpreted something?

9. Fig. 4 – A consistent y-axis scale would aid comparison of volume expansion across species and food types.

Reviewer #3

(Remarks to the Author)

The author's primary goal is to test "Liem's Hypothesis", which states that morphological specialization can be associated with increased dietary flexibility". This is a tricky hypothesis to test and, the authors specifically "evaluate whether and how mouth cavity expansion is different in predatory vs algae-eating species". Their secondary goal is to "infer how the cranial expansion system is functionally tuned to this type of food, and how modifications may adversely affect the fish's capacity for powerful suction generation as observed in piscivores". This experimental study is carried using three species of African rift lake cichlids, the very group Karel Liem used to found his hypothesis, with dietary scopes ranging from algae scraper, via algae picker, to piscivory. The general premise of the study is to use 3D time-varying kinematics of the head during suction feeding and biting to generate internal 3D flow profiles of the oral, buccal, and pharyngeal head regions to assess if the piscivore, which is specialized on capturing errant food, has faster and more rostrocaudally-sequenced flow generation than the algae feeders. In general, I am excited about this study, but I have a couple of concerns and some suggestions to improve the quality of the manuscript.

Concern.

1. Approach. This study takes a purely geometrical approach to predicting, and not demonstrating, flow and its rate of progression the mouth, hyoid and branchial cavities of expanding fish heads during suction generation. The specific concern is that the rates and progressions of flow fields are not validated directly, e.g. using implanted pressure probes (which I acknowledge would be difficult due to the small specimen sizes) nor indirectly, using videofluoroscopic imaging of neutrally buoyant particles (Provini et al., E-life 2022), or the "dynamic endocast" approach (Camp et al., PNAS 2015) despite that methods are available and the corresponding author is experienced with these. Moreover, there is no morphological assessment, or citations of same, to indicate the presence or absence of potential mechanical obstructions of flow within the cavities of the three cichlid species studied. It is well known that teleost fishes have integumental "valves" that are important in orchestrating the progression of the suction feeding event. However, prior studies were almost all based on suction feeding specialists, wherefore it remains unclear to which extent these valves change in prominence and/or function across a gradient of species from suction to biting specialist, as studied here. Therefore, without either direct assessment of flow, or morphological assessment of flow prevention, this study cannot per se provide empirical evidence of flow patterns. However, I acknowledge that this study is a great hypothesis generator for future flow studies, but the wording here and there should be altered (L. 327, L364, to acknowledge the aforementioned shortcoming.

At the end of the introduction (L. 82-85), you present two new intentions and it is not at all clear how this was done, so you may want to flesh the approach out in the introduction and stats section.

2. Statistics. L80-81. This is the goal of the study. In the stats section it is not clear how this is tested. The stats section (starting L 208) should carefully specify which part of the statistical model is actually used to evaluate the HYPOTHESIS that biters have a different flow profile for free food than suction feeding specialists. NOTE: This study is not setting up a specific hypothesis to be used to test Liem's Paradox, but I guess that is OK.

L214. Would the interaction-term between food type and species not be the factor that is relevant for assessing if different species adjust differently to free food? In either case, it is critical that you highlight to the reader which component(s) of your stats design allow(s) you to test your aims: "evaluate whether and how mouth cavity expansion is different in predatory vs algae-eating species" and "infer how the cranial expansion system is functionally tuned to this type of food, and how modifications may adversely affect the fish's capacity for powerful suction generation as observed in piscivores".

L326: The statistical model you use needs to capture this focus and claim. At least, you need to adjust the stats paragraph in the methods section. I am thinking that you maybe could use SPM (Statistical Parameter Mapping) to compare the time-varying shape of your head volume change patterns in Figure 4?

Suggestions.

Introduction: The introductions should be re-sequenced, in order to capture the attention of the journal's broad readership. For sure, this paper is on cichlid feeding kinematics, and that would be a fine place to start if this manuscript was for a fish-focused journal. But is your scope not way broader? i.e., Liem's hypothesis? (L25-26). The data support an oft speculated but rarely tested trade-off between biting and suction strategies in the procurement stage of the feeding behavioral repertoire in fishes. I would suggest you start with a short paragraph on the more fundamental motivation(s) for your study, then treat the model system and its appropriateness for carrying out the testing (being the very system Karel Liem used). This would also make for a stronger parallel sequencing of your introduction with the discussion (e.g. L. 318-319, initiating the discussion with the conundrum which, if it is at the heart of this study should be telegraphed in the introduction!). Similarly, in paragraph 2. Again, this is a broader principle, not exclusive to African Rift Lake cichlids (ARLC). So, the topic sentence could talk about this principle and then use "for instance" or "for example" when referring to ARLC's. I really like the way Liem's idea is presented as a chronological account in paragraph 2, but I would suggest moving this up to help set up the study around Liem's Paradox (and cite the original 1980 paper!).

Minor suggestions

L4. "transport detached algae after it has been scraped or picked off, or procured"?

L7. Important to highlight the food type, (free) rather than its shape (round).

L55. I don't know that this is correct. See literature on angelfishes, and recent papers on surgeonfishes.

L59. Respiration also (swap word sequence)
L60. Combine to buccopharyngeal or "mouth" cavity
L75. Should be plural: "morphologies and feeding behaviors" right?
L99. Other ingredients are not necessary to list here, are they?
L109-110. Why? Satiation? Cite Sass and Motta?
L144 should be presented before L141, as the latter otherwise does not make sense.
L154. Symbol needs definition and if this is PI, please replace with symbol used in line 197.
L156. Start with "Time-varying profiles for the six kinematics variables"?
L170. Please provide a citation or other indication of how this function can be obtained for evaluation.
L201. Not quite a sentence.
L350-353. So, context-dependent?

Version 1:

Reviewer comments:

Reviewer #2

(Remarks to the Author)

I appreciate the authors' rigorous and thoughtful engagement with all of the reviews. Their responses have been genuinely helpful in better understanding this work, and I wish more authors provided responses like this!

This remains an important, novel, and well-supported study – I have no further concerns.

Based on their edits and responses, I am satisfied that the interpretations of efficiency and adaptive value are appropriately framed as reasonable, biomechanically-supported hypotheses.

The re-analysis of volume calculations presented in the rebuttal has addressed my concerns about volume calculations. I empathize with the difficulties of normalizing expansion volume – the method they have chosen is reasonable and well-explained, and I don't have a better one to suggest at the moment.

Below are two Minor typos that I spotted in case the authors find this is helpful -- these do not in any way impact the scientific quality or understanding of the work.

Line 43 – consider rewording to "If the preferred food is abundant, fish specialized in other food types will also consume it..."

Line 698 – should this be "tablets" instead of "tables"?

Reviewer #3

(Remarks to the Author)

I congratulate the authors on an effective revision of what I already thought was a nice manuscript.

1. Apologies for misinterpreting figure 4 as flow when I initially skimmed your MS. The morning coffee evidently had not taken effect and, unfortunately, that sentence persisted from my notes into my review of your MS.
2. I think it would be worthwhile to include your comparative analysis of volume estimation methods in your supplemental materials. Given your findings from that analysis, it would be pertinent to include a sentence ala: "We compared volumetric expansion methods and found a relatively small (4-14%) overestimation by the half-ellipsoid, as compared to the multi-fustrum method (see SI Fig 2). The trend was such that the differences between species would marginally increase using ellipse-based normalization".

We sincerely thank the reviewers for their excellent, constructive suggestions on how to improve our article. We greatly appreciate your time and effort in writing these reviews. We apologise for the delay we had in preparing this revision.

Note that the abstract had to be shortened significantly from 200 to 150 words, as requested by the editor.

Below, the original review reports are given, with our replies printed in blue.

Reviewers' comments:

Reviewer #1 (Remarks to the Author):

This study investigates suction mechanics in three cichlid species from Lake Malawi differing in their head and mouth morphology, and foraging preferences. The authors measured mouth cavity expansion with volumetric analyses, suction capacity and three-dimensional kinematics, and provide qualitative descriptions of foraging behaviors.

The study finds interesting differences between piscivorous and alga-eaters in their feeding strategies, suggesting that the supposedly less efficient suction kinematics of algae feeders could have some benefits for catching smaller food particles, and the absence of waves of expansion and compression of the head while feeding in algae feeders, instead showing synchronous expansion patterns. AN interesting result is that all species show the capacity to modulate their feeding kinematics in function of food type.

Overall, I find the study interesting and well done.

>>Thank you!

Minor comments

-In the introduction and Figure 1 there are some morphological features of the three species analyzed, but this is not mentioned in the results and not really discussed. Either this is taken from a previous study, which I do not see clear, or should be more formally presented here.

>> We followed the reviewer's suggestion to include a description in the results section of the morphological differences between the species based on Figure 1. This is now the first paragraph of the Results.

-Related to the previous point, this study evaluates kinematic differences in feeding behavior, but it could be interesting to also see some discussion on the morphological differences in mouth and head morphology, other than volumes, that could relate to the preferred behaviors. One example could be mouth size and its relation to degree of mouth opening while suction. Another interesting aspect could be the elongation of the head and its relation to suction power.

>> We fully agree it would be interesting to link the kinematic differences, particularly the characteristics of the expansion, back to the morphology. The quest for the anatomical or constructional basis of the observed results is the topic of the final paragraph of the discussion, before the conclusions. As mentioned in that paragraph, this is a complex topic involving coupled

kinetics of the many mobile components of the head. Few studies have performed such kinetic modelling. We feel this is beyond the scope of the current study. In the revised version, we include an additional reference to a study that links the head morphology, including differences in hyoid angle related to head width as in our study, to timing and magnitude of head expansion during suction feeding: De Visser, J. and Barel, C. D. N. (1998). The expansion apparatus in fish heads, a 3-D kinetic deduction. *Neth. J. Zool.* 48, 361–395. The narrow, elongated shape of the cichlid's head, as suggested by the reviewer, is also a factor of importance in these models. The link to power, provided by the axial muscle system, however, is not a morphological aspect we can easily integrate (e.g not visible in Fig. 1). Lower jaw morphology is indeed reflected in the observed gape size, as shown in Fig. 3a. However, this seems self-evident; therefore, we believe it does not need further discussion. We hope the final paragraph inspires follow-up studies to deal with this interesting topic in more detail. The new paragraph reads:

"Morphology

The three species studied, for which the results will be reported in order of increasing specialisation in algae feeding, differed in head shape and skeletal morphology (Fig. 1). The most notable morphological differences were evident in the head's width-to-length aspect ratio (maximum head width divided by jaw-tip-to-opercular-tip length), which increased from 0.36 in R. sp. 'Chilingali' (Fig. 1a), to 0.61 in C. saulosi (Fig. 1b), and reached 0.67 in L. trewavasae (Fig. 1c). The orientation of the lower jaw rami (refer to dentary and articular labels in Fig. 1) from the symphysis to the quadrate joint, as viewed from a ventral perspective, increased from about 10° (Fig. 1a), to 30° (Fig. 1b), and culminated at 45° (Fig. 1c) among the species in our series. The angle between the ceratohyals and the midsagittal plane increased from approximately 15° (Fig. 1a), to 30° (Fig. 1b), and to 50° (Fig. 1c) across the three species. Oral tooth shape ranged from relatively long and sharp unicuspid teeth (Fig. 1a), to smaller and more densely packed bicuspid and tricuspid teeth (Fig. 1b), and to multiple layers of tricuspid teeth on a mediolaterally straight jaw (Fig. 1c)."

Reviewer #2 (Remarks to the Author):

Summary

This study investigates the long-standing question of how the feeding apparatus of fish has contributed to (or restricted) their trophic and morphological diversity. Specifically, do specializations for biting or scraping create trade-offs with suction feeding performance? This is important for understanding how and why evolutionary shifts occur in adaptive radiations – of which African cichlids are a famous example.

The authors compare feeding in 3 African cichlid species (a piscivore, an algae picker, and an algae scraper) performing suction feeding (on shrimp) and biting/scraping (on algae). For each species and food type they measure the kinematics of dorsoventral and lateral mouth expansion and the change in cross-sectional area along the head and throughout the behaviour. The main conclusions are:

1. Algae eaters use a synchronous—rather than anterior-to-posterior—mouth expansion
2. Synchronous expansion is adaptive for algae eating
3. Synchronous expansion precludes powerful suction, forming a trade-off between powerful and efficient feeding for cichlid fishes and ultimately shaping trophic niches and diversification

Overall Impression

The first conclusion (synchronous expansion throughout the head in the two algae-eating species) is well-supported by the kinematic data, novel, and potentially transformative for studying the mechanics and ecology of fish feeding. Prior to this, an anterior-to-posterior wave of expansion was considered a fundamental and universal element of suction feeding in fish. By demonstrating this alternative strategy for suction feeding, this study presents two key findings that have been largely overlooked in the field:

1. Slower, smaller-magnitude suction feeding is not necessarily “low performance”, but can be a successful strategy for consuming non-evasive food over longer durations. I agree wholeheartedly with the authors’ conclusion “Suction kinematics cannot be simply deemed either efficient or inefficient as it depends on the type of food that is considered” (p17, line 427). This raises interesting and important questions about when power—as opposed to work or efficiency—is important for suction feeding, and how muscles are employed for these different feeding strategies.
2. The kinematics of individual elements of the feeding apparatus need to be understood in the context of the volumetric expansion of the entire mouth cavity. I found it fascinating that an anterior-to-posterior wave of expansion could be interpreted from the linear kinematics of the two algae feeders, but not the measurements of mouth cross-sectional area. As the authors point out, this suggests synchronous expansion could be quite common but undetected in suction feeding fish (p17, lines 408-414).

I very much hope the field will embrace these findings and pursue the new avenues they open up.

>>Thank you!

The remaining two conclusions (on the adaptive advantages of algae-eating and the trade-off between powerful and efficient feeding) are plausible, but not strongly supported.

The authors made no measurements of feeding power, efficiency, or any type of performance; instead these conclusions are inferred from the kinematics (p7 line 83). But to my knowledge none of these can be accurately predicted solely from kinematics. For example, Van Wassenbergh et al., 2006 demonstrates the variable relationship between suction performance, volume, pressure, and morphology across different fishes.

The proposed explanations for how synchronous expansion is adaptive for algae eating are reasonable, but not tested or supported with any data. Can the authors refute the alternative hypothesis: synchronous expansion is sufficient for algae eating, but no more adaptive/advantageous than an anterior-to-posterior wave of expansion?

>> We fully agree that the explanations provided in the Discussion, are not tested. However, we think we handled this correctly by posing these explanations as hypotheses, which implies that alternatives cannot be excluded by our data. For example, we hypothesise that the low-amplitude, synchronous expansion is an adaptive characteristic of pump-filter-feeding fish like the algae-eating cichlids based on arguments for why this may provide a functional advantage. We hope the hypotheses make sense from a biomechanical point of view, which seems supported by the favourable reviews.

The piscivore’s capacity for powerful suction may not be the reason it is incapable of efficiently feeding on algae. First, we do not know how powerful or (in)efficient the piscivore actually is (see point above). This is a general problem in the field of how to compare feeding power across different

species (e.g., Camp and Brainerd, 2022) and how to define a suction feeder as almost all fish use some combination of suction and ram (Longo et al., 2015). Second, even if we assume the piscivore is both powerful and inefficient at algae feeding, can the authors refute alternative hypotheses for its poor ability to feed on algae, such as its morphology or reliance on ram or body posture? For example, how might a powerful, piscivorous suction feeder with short-jaws and relatively little body ram—such as a bluegill sunfish—perform on algae?

>>We understand the difficulty of concluding on efficiency, as this cannot be measured easily for suction feeding. Still, we provided a few novel ideas of how adaptation for powerful suction may interfere with algae feeding. As argued in the manuscript, the ability to create a low-momentum outflow to aid small food filtration while still creating suction in fast successive bouts is not something we see a specialist piscivore capable of. Our piscivore reduced its expansion significantly (Fig. 4a) but still showed the fastest expansion and largest opercular abduction (Fig. 3e,f). A similar result can be expected for a bluegill sunfish.

The work is described in sufficient detail that it could be replicated, and I am pleased to see the authors have already prepared to make the video data openly available on FigShare. The statistical analysis seems appropriate to me, although this is not an area in which I am an expert.

Thank you!

Specific Comments

1. The impact of different head shapes, volumes, and volume calculation methods on the resulting expansion area and volume data needs to be discussed. As all the data are reported relative to resting head volume, this value is quite important. Please address the following questions in the methods or discussion, or provide references if this is already addressed in the literature.

A) Why was resting head volume calculated from only 3 measurements and modelling the head as a half ellipse (p9 lines 151-155)? It seems a more accurate measure could have been made from instantaneous head volume calculations at a frame before the onset of feeding (lines 174-206).

>> This is a valid remark, and admittedly it would have been more logical to use the same method as for the volumetric analyses. The question is whether the simple method (half ellipsoid volume) differs significantly from the more advanced, labour-intensive method (multi-ellipse volume) to necessitate rescaling all data and figures and redoing the statistics. We therefore analysed both methods for each species studied and found a relatively small and consistent (across species) overestimation by the half-ellipsoid method (see image below). The trend is such that the differences between species would marginally increase using ellipse-based normalisation. We hope this reassures that a sensible normalisation has been used, and can be retained in the manuscript. In the revised version, we added a reference for the shape approximation by a half ellipsoid: *“To calculate the resting volume of each individual, the shape of the head was approximated as a half ellipsoid, since this shape generally closely resembles the head of a fish in an unexpanded state (Van Wassenbergh et al. 2015).”*

half ellipsoid volume =	27906	24873	30158
multi-frustum volume =	24505	22933	29090
difference (%) =	+14%	+8%	+4%

B) The two algae-eating species had broader and higher-volume heads; may this have impacted the normalised volume expansion? Could this method of measuring volume change from external head profiles introduce any bias when comparing species with different head shapes, e.g., the more slender piscivore and the more robust algae-eaters?

>> We understand this remark but don't immediately see a better alternative. It is logical to scale head volume expansion by the initial (resting state) head volume. The point of the reviewer is probably that some parts of the head, like the neurocranium and jaw adductors, do not directly participate in the expansion process, so it is not ideal to include these in the normalization. For example, a dorsal outgrowth of the braincase would lower the normalised expansions of the same buccal cavity, if scaled by total head volume. Still, the fact that we scale by head volume already removes irrelevant body parts compared to using normalization to total body mass or body volume. Measurement of the initial volume of the internal mouth cavity may be better, but this is difficult to estimate and prone to error. Most important in our view, apart from making the metric dimensionless, is that the meaning is clear. From an evolutionary perspective, the expansion performance for a given head volume is meaningful. We hope the reviewer can connect to this view.

2. Personally, I feel a more detailed analysis of the interaction of kinematics and morphology, which is mentioned briefly (p17 lines 415-421), would add substantial value to this study. How much the different head shapes of these species could be impacting their kinematics and expansion? There are some really exciting questions here and the authors have already detailed 3D morphological data (Fig. 1). But I leave it to the authors to decide if such detail is within the scope of their study. It is a testament to the novelty and impact of their work that it already has me excited about new studies and questions!

>> We fully agree that a logical extension of our study would be to investigate the anatomical or constructional basis of the observed kinematic results. This is also what reviewer 1 suggested. This would be challenging and demanding work, as explained in response to reviewer 1's comment. An additional reference is added to the revised manuscript to illustrate this: De Visser & Barel 1998 on

the hyoid-suspensorium linkage kinetics. We, therefore, decided to leave this out of the scope of the current study and keep the focus on the expansion kinematics.

3. Why wasn't the maxilla tracked in ventral views—was it not clearly/consistently visible? Lateral abduction of the oral jaws seems like it is also contributing to mouth volume change, and getting this rostralmost region would help with tracking anterior-posterior timing. A measure of mouth width would have also allowed you to calculate the area of the mouth opening, which can be quite important for suction flows.

>> That may have been possible, but the maxilla is very close to the suspensorium landmarks placed on the very front of the suspensorium. This already reflects the abduction of the (lower) jaws. Maxilla abduction is included in the contour tracings and, therefore, the volume change graphs (Fig. 4). The area of the mouth opening is difficult to estimate accurately: it is non-planar in the piscivore (Fig. 2b), and from the ventral views, it is difficult to decide where on its arched lower jaw the mouth opening width should best be measured. The fleshy mouth of the algae scraper makes it difficult to pinpoint the internal border of the mouth opening from a lateral view.

4. P6, line 50 – Do they really have conflicting morphological demands, given the content of the next two sentences? Consider re-wording this.

There is evidence for conflicts and trade-offs between species using predominantly suction and those specialising in biting, despite that these functions can be combined to some extent. We added references to support the first part of the sentence on the 'conflicting morphological demands, namely Barel 1983; De Visser & Barel 1996; De Schepper et al. 2008; Burns et al. 2024.)

5. P6 lines 60-67; 82 – Please define what specific measurement is meant by 'mouth expansion' and is it the same this study (line 82) and Martinez et al 2018 (lines 60-67)?

>> We now specify that the Martinez et al. 2018 study measured in 2D and was based on lateral-view high-speed videos, while this study measures expansion in 3D. Neither strictly measures the expansion of the "cavity," but we believe the assumption that the cavity expands along with the external landmarks is quite robust.

6. P8 lines 139-147– Please explain how was the y-axis of the head reference frame defined? And how was the constant angle determined?

>> This information was added to this paragraph. Our x-axis was taken parallel to the line from the dorsal attachment of the pectoral fin to the mouth centre. The y-axis is orthogonal to this x-axis.

7. P11, line 237 – This is the first and only mention of pectoral girdle retraction: was it measured in any species? Did it occur in any species? Please provide more detail or remove this reference.

>> It was not measured, so we fully agree this sentence should better be deleted.

8. P12, lines 253-259 – I'm surprised that suspensorium abduction was significantly different between food types for the piscivore (as these lines look very similar in Fig. 3D)...but then hind

operculum was NOT different between food types for the picker (as these lines look quite different in Fig. 3F). Have I misinterpreted something?

>> It is true that less than 10% difference can be observed in the peak of the mean expansion profile Fig. 3D for the piscivore. However, there can be some difference in the mean peak values of the individuals due to different timings of the individual peaks. The statistics also accounts for individual (paired test), which cannot be viewed in the figure. The same applies to the algae picker's hind operculum. There is a tendency towards a significant difference ($p = 0.01$) for this variable.

9. Fig. 4 – A consistent y-axis scale would aid comparison of volume expansion across species and food types.

>> We followed this suggestion and changed the y-axes of the normalized volume changes so that expansions can be compared easily 'on sight'.

As requested by the editors, we here include the revised figure in the response file:

Reviewer #3 (Remarks to the Author):

The author's primary goal is to test "Liem's Hypothesis", which states that morphological specialization can be associated with increased dietary flexibility". This a tricky hypothesis to test and, the authors specifically "evaluate whether and how mouth cavity expansion is different in predatory vs algae-eating species". Their secondary goal is to "infer how the cranial expansion system is functionally tuned to this type of food, and how modifications may adversely affect the fish's capacity for powerful suction generation as observed in piscivores". This experimental study is carried using three species of African rift lake cichlids, the very group Karel Liem used to found his hypothesis, with dietary scopes ranging from algae scraper, via algae picker, to piscivory. The general premise of the study is to use 3D time-varying kinematics of the head during suction feeding and biting to generate internal 3D flow profiles of the oral, buccal, and pharyngeal head regions to assess if the piscivore, which is specialized on capturing errant food, has faster and more rostrocaudally-sequenced flow generation than the algae feeders. In general, I am excited about this study, but I have a couple of concerns and some suggestions to improve the quality of the manuscript.

>> Thank you!

Concern.

1. Approach. This study takes a purely geometrical approach to predicting, and not demonstrating, flow and its rate of progression the mouth, hyoid and branchial cavities of expanding fish heads during suction generation. The specific concern is that the rates and progressions of flow fields are not validated directly, e.g. using implanted pressure probes (which I acknowledge would be difficult due to the small specimen sizes) nor indirectly, using videofluoroscopic imaging of neutrally buoyant particles (Provini et al., E-life 2022), or the "dynamic endocast" approach (Camp et al., PNAS 2015) despite that methods are available and the corresponding author is experienced with these. Moreover, there is no morphological assessment, or citations of same, to indicate the presence or absence of potential mechanical obstructions of flow within the cavities of the three cichlid species studied. It is well known that teleost fishes have integumental "valves" that are important in orchestrating the progression of the suction feeding event. However, prior studies are were almost all based on suction feeding specialists, wherefore it remains unclear to which extent these valves change in prominence and/or function across a gradient of species from suction to biting specialist, as studied here. Therefore, without either direct assessment of flow, or morphological assessment of flow prevention, this study cannot per se provide empirical evidence of flow patterns. However, I acknowledge that this study is a great hypothesis generator for future flow studies, but the wording here and there should be altered (L. 327, L364, to acknowledge the aforementioned shortcoming.

>> It seems the reviewer has made a mix-up regarding our study's data. We concentrated specifically on the kinematics of the cranial elements and the volumetrics, but contrary to what the reviewer wrote, we have not calculated water flows or flow fields. The approach for volumetrics was validated for fish heads in the original description of the technique by Drost and van den Boogaart (1985).

We didn't explore deriving flow velocities from the local volume changes. Technically, this would be possible, based on the principle of continuity and after measuring the initial internal cross-sectional

areas of the buccal cavity, though only for periods when either only the mouth or the opercular valves are open. Our group did such flow velocity calculations before on different fish, but here, we felt that the differences in expansion kinematics were sufficient to evaluate the species differences in the way they feed. The mentioned experimental techniques would be extremely difficult, if not impossible, given the small size of the species from our study.

Although buccal expansion drives suction flows and is, therefore, interesting on its own, we agree that water flow characteristics matter eventually for food capture. We therefore included a new paragraph acknowledging that our study is limited by not having measured or calculated flow velocities. We agree that our study's important value is presenting new hypotheses for follow-up studies, and also highlight this message in the new paragraph. The new paragraph reads:

“The implications of the observed variation in cranial expansion kinematics, along with the potential variations in the morphology of the buccal cavity, on the dynamics and patterns of water flow require further investigation. In the scope of the hypothesised trade-off between high-power suction and small-food suction, it would be informative to quantify the outflow properties as well, e.g. flow speed and momentum at the opercular exit. As mentioned above, the mechanical and morphological demands of these two tasks are vastly different. Quantifying the kinematics of the gill arches would also yield valuable insights. These arches are known to spread considerably after powerful suction (van Leeuwen 1984) and may, therefore, compromise the functionality of small-food sieving. Several experimental approaches exist to gain such insights (e.g., Higham et al. 2006; Weller et al. 2020; Provini et al. 2022). However, for Malawi cichlids, the generally small head size may be a limiting factor.”

Line 364 was toned down, changing from ‘it implies...’ to ‘it may imply a reduced outflow speed and momentum of the water’.

At the end of the introduction (L. 82-85), you present two new intentions and it is not at all clear how this was done, so you may want to flesh the approach out in the introduction and stats section.

>> We agree this was confusing, as these ‘intentions’ were purely to discuss these aspects rather than to tackle this directly by specific measurements. We changed the sentence from ‘From this we will infer...’ to ‘From this we will argue how the cranial expansion system may be functionally tuned...’

2. Statistics. L80-81. This is the goal of the study. In the stats section it is not clear how this is tested. The stats section (starting L 208) should carefully specify which part of the statistical model is actually used to evaluate the HYPOTHESIS that biters have a different flow profile for free food than suction feeding specialists. NOTE: This study is not setting up a specific hypothesis to be used to test Liem's Paradox, but I guess that is OK.

L214. Would the interaction-term between food type and species not be the factor that is relevant for assessing if different species adjust differently to free food? In either case, it is critical that you highlight to the reader which component(s) of your stats design allow(s) you to test your aims: “evaluate whether and how mouth cavity expansion is different in predatory vs algae-eating species” and “infer how the cranial expansion system is functionally tuned to this type of food, and how

modifications may adversely affect the fish's capacity for powerful suction generation as observed in piscivores".

L326: The statistical model you use needs to capture this focus and claim. At least, you need to adjust the stats paragraph in the methods section. I am thinking that you maybe could use SPM (Statistical Parameter Mapping) to compare the time-varying shape of your head volume change patterns in Figure 4?

>> We agree the section on statistics would benefit from referring to the aims and hypotheses. We added the following text to this section: "To determine how suction-feeding differs between piscivorous and algae-eating cichlids, we tested the hypotheses that the three species differ in their cranial kinematics. Additionally, we also tested whether the two food types cause changes in kinematics, and whether the species respond differently to changes in food type (i.e., the interaction effect between 'food type' and 'species'). The analysis was performed on the anatomical landmark-based kinematic data (Fig. 2b)." The interaction effect is certainly interesting, but the focus on performance trade-offs between morphologies calls for a test on differences between species as a main focus. This is reflected in the newly added text shown above.

As mentioned earlier, the phrase 'infer how the cranial system is functionally tuned...' was reformulated to clarify that this will not be a result of a statistical test. We apologize for the confusion this has caused.

We thank the reviewer for suggesting statistical testing methods for multidimensional maps using Statistical Parameter Mapping. However, as the sample size for the analysis is smaller than the landmark-based kinematics and there is some overlap (e.g. the decreased expansion amplitudes in the algae eaters), we opted to present these results (Fig. 4) descriptively, without a statistical test. The patterns described seem clear enough (visibly) to be presented in this way.

Suggestions.

Introduction: The introductions should be re-sequenced, in order to capture the attention of the journal's broad readership. For sure, this paper is on cichlid feeding kinematics, and that would be a fine place to start if this manuscript was for a fish-focused journal. But is your scope not way broader? i.e., Liem's hypothesis? (L25-26). The data support an oft speculated but rarely tested trade-off between biting and suction strategies in the procurement stage of the feeding behavioral repertoire in fishes. I would suggest you start with a short paragraph on the more fundamental motivation(s) for your study, then treat the model system and its appropriateness for carrying out the testing (being the very system Karel Liem used). This would also make for a stronger parallel sequencing of your introduction with the discussion (e.g. L. 318-319, initiating the discussion with the conundrum which, if it is at the heart of this study should be telegraphed in the introduction!). Similarly, in paragraph 2. Again, this is a broader principle, not exclusive to African Rift Lake cichlids (ARLC). So, the topic sentence could talk about this principle and then use "for instance" or "for example" when referring to ARLC's. I really like the way Liem's idea is presented as a chronological account in paragraph 2, but I would suggest moving this up to help set up the study around Liem's Paradox (and cite the original 1980 paper!).

>> Thank you for this excellent suggestion. We followed it by adding a new paragraph with a broad scope on trade-offs and their importance in ecology and evolution. Although Liem's paradox is

connected to it, our study is essentially about identifying trade-offs, and not directly about Liem's paradox (as the reviewer noted him/herself in writing in remark 2 that we are not testing Liem's paradox). This is also the message of the first paragraph of the discussion: Liem's paradox is intriguing in cichlids, but it does not change the fact that functional trade-offs are involved and that these are critical for diversification.

The new kick-off paragraph in the introduction now reads:

"Understanding functional trade-offs plays a central role in organismal and evolutionary biology (Garland et al., 2022). These trade-offs derive from conflicting functional demands, implying that adaptations for one specific task can lead to reduced performance in another. They can drive the evolution of specialization in which species occupy narrow ecological niches where they excel (Vamosi et al. 2014; Østman et al. 2014). Specialization is common in adaptive radiations, where ancestral species diversify into various forms, each adapted to a different ecological role (e.g., Salzburger et al., 2014; De León et al. 2012). However, the trade-offs accompanying such specialization can also limit a species' ability to adapt to new environments or shifts in ecological conditions (e.g., Willy & Van Buskirk 2021). Therefore, identifying conflicting demands on organisms' morphology, physiology, mechanics, or behaviour is crucial for understanding ecological niches and evolutionary processes."

Next we connect it to the existing intro on cichlids by: "An important model system in the study of the role of functional trade-offs in the evolution of an adaptive radiation are cichlids (Hulsey & Garcia de León 2005; Ronco et al. 2019; Burress et al., 2020; Burress & Muñoz 2023)."

Minor suggestions

L4. "transport detached algae after it has been scraped or picked off, or procured"?

>> We like this suggestion, but unfortunately the word limit prevents us from adding this phrase.

L7. Important to highlight the food type, (free) rather than its shape (round).

>> Indeed. We changed this as suggested, naming the food type 'free-sinking'.

L55. I don't know that this is correct. See literature on angelfishes, and recent papers on surgeonfishes.

>> We think the majority is still about jaws and biting, but it does warrant a 'but see'. We now cite Bouton et al. (1998), Rupp & Hulsey (2014), and Mihalitsis & Wainwright (2024) to illustrate a few studies that also considered the suction apparatus.

L59. Respiration also (swap word sequence)

L60. Combine to buccopharyngeal or "mouth" cavity

L75. Should be plural: "morphologies and feeding behaviors" right?

>>these three remarks are fixed as suggested.

L99. Other ingredients are not necessary to list here, are they?

>> Indeed, these can be found on the product if needed. We removed these 'other ingredients'.

L109-110. Why? Satiation? Cite Sass and Motta?

>> Indeed to avoid satiation. We added the reference.

L144 should be presented before L141, as the latter otherwise does not make sense.

>> Indeed, we switched the order.

L154. Symbol needs definition and if this is π , please replace with symbol used in line 197.

>> the symbol was changed to π .

L156. Start with “Time-varying profiles for the six kinematics variables”?

>>changed as suggested.

L170. Please provide a citation or other indication of how this function can be obtained for evaluation.

>>We can't find a function in L170. However, all software used will be referenced in an accompanying 'reporting summary' document. For software this will be: “Kinematic and volumetric data calculations Microsoft Excel version 365 (2024); Statistics: R version 4.0.3; Statistics R-packages: lmerTest package, version 3.1.3; pbkrtest package, version 0.5.1; emmeans package, version 1.7.4-1”

L201. Not quite a sentence.

>> this formed a sentence together with L199 and the formula on L200.

L350-353. So, context-dependent?

>> Yes, but the context, feeding on algae, was already mentioned so it doesn't seem necessary to repeat this.

Thank you very much for these insightful reviews!

Thank you for the appreciation and suggestions. Below, the original review reports are given, with our replies printed in blue.

Reviewer #2 (Remarks to the Author):

I appreciate the authors' rigorous and thoughtful engagement with all of the reviews. Their responses have been genuinely helpful in better understanding this work, and I wish more authors provided responses like this! This remains an important, novel, and well-supported study – I have no further concerns. Based on their edits and responses, I am satisfied that the interpretations of efficiency and adaptive value are appropriately framed as reasonable, biomechanically-supported hypotheses. The re-analysis of volume calculations presented in the rebuttal has addressed my concerns about volume calculations. I empathize with the difficulties of normalizing expansion volume – the method they have chosen is reasonable and well-explained, and I don't have a better one to suggest at the moment.

>>Thank you!

Below are two Minor typos that I spotted in case the authors find this is helpful -- these do not in any way impact the scientific quality or understanding of the work.

Line 43 – consider rewording to “If the preferred food is abundant, fish specialized in other food types will also consume it...”

Line 698 – should this be “tablets” instead of “tables”?

>>We changed these sentences as suggested.

Reviewer #3 (Remarks to the Author):

I congratulate the authors on an effective revision of what I already thought was a nice manuscript.

1. Apologies for misinterpreting figure 4 as flow when I initially skimmed your MS. The morning coffee evidently had not taken effect and, unfortunately, that sentence persisted from my notes into my review of your MS.

>>No problem.

2. I think it would be worthwhile to include your comparative analysis of volume estimation methods in your supplemental materials. Given your findings from that analysis, it would be pertinent to include a sentence ala: "We compared volumetric expansion methods and found a relatively small (4-14%) overestimation by the half-ellipsoid, as compared to the multi-fustrum method (see SI Fig 2). The trend was such that the differences between species would marginally increase using ellipse-based normalization".

>> As suggested, we now included this comparison as a supplementary figure (new Supplementary Figure 1), and added a sentence to the main manuscript that summarized the result.